# Eagle 2.5: Boosting Long-Context Post-Training for Frontier Vision-Language Models

**Guo Chen[1,2]\*, Zhiqi Li[1,2], Shihao Wang[3,2], Jindong Jiang[2], Yicheng Liu[1], Lidong Lu[1], De-An Huang[2], Wonmin Byeon[2], Matthieu Le[2], Max Ehrlich[2], Tong Lu[1†], Limin Wang[1†], Bryan Catanzaro[2], Jan Kautz[2], Andrew Tao[2], Zhiding Yu[2†], Guilin Liu[2†]**

[1]Nanjing University, [2]NVIDIA, [3]Hong Kong Polytechnic University

chenguo1177@gmail.com

## Abstract

We introduce Eagle2.5, a frontier vision-language model (VLM) for long-context multimodal learning. Our work addresses the challenges in long video comprehension and high-resolution image understanding, introducing a generalist framework for both tasks. The proposed training framework incorporates Automatic Degrade Sampling and Image Area Preservation, two techniques that preserve contextual integrity and visual details. The framework also includes numerous efficiency optimizations in the pipeline for long-context data training. Finally, we propose Eagle-Video-110K, a novel dataset that integrates both story-level and clip-level annotations, facilitating long-video understanding. Eagle2.5 demonstrates substantial improvements on long-context multimodal benchmarks, providing a robust solution to the limitations of existing VLMs. Notably, our best model Eagle2.5-8B achieves 72.4% on Video-MME with 512 input frames, matching the results of top-tier commercial model such as GPT-4o and large-scale open-source models like Qwen2.5-VL-72B and InternVL2.5-78B.

## 1 Introduction

Despite the significant advances in multimodal learning [4, 16, 52, 53, 97], many vision-language models (VLMs) remain focused on short-context tasks, with long-context understanding under-explored. This gap is particularly evident in both long video comprehension and high-resolution image/video understanding, where the processing of extended visual contexts remains an open challenge. Such extended contexts encompass multiple images, extended video sequences, high-resolution media, or combinations thereof. However, the development of long-context VLMs is still in its early stages, hindered by fundamental challenges in dataset construction, architecture design, training strategies, and computation/memory bottlenecks.

To enable long-context visual understanding, several approaches have been proposed to address the challenge of processing extended visual inputs by designing specialized compression or selection modules [52, 84, 57, 45, 43, 117, 103, 53]. While these methods effectively circumvent the need to extend the context length of VLMs, they often introduce additional computational overhead or capacity limitations, potentially constraining model performance. A promising research direction is to extend the context length of LLMs to enable native long-context understanding. While prior studies [110, 126, 85] have explored this direction, challenges and key limitations still remain. First, the performance of existing methods is often suboptimal, generally falling behind proprietary models. Second, these approaches struggle to achieve consistent improvements as the amount of visual input

---

*Work done during an internship at NVIDIA.

†Co-corresponding authors.

increases. Lastly, the optimal training strategies for state-of-the-art long-context VLMs remain unclear, given the complex interplay of factors such as training strategies and data recipes.

To this end, we present Eagle 2.5, a versatile multimodal model designed to efficiently process extensive contextual information. Unlike models solely optimized for handling long multimodal sequences without improving performance, Eagle-2.5 benefits from increased input length, leading to consistent performance gains besides merely accommodating longer inputs. As shown in Fig. 1, our model achieves superior context coverage and exhibits consistent performance scaling with increasing frame counts. Notably, it attains competitive results compared to larger models such as GPT-4o [77] and Qwen2.5-VL-72B [5], while maintaining a significantly smaller parameter footprint.

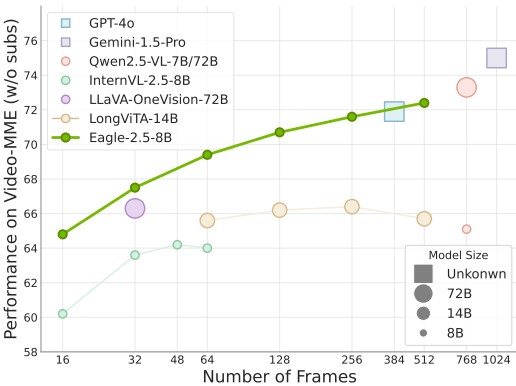

Figure 1: **Performance comparison of Eagle2.5 with leading vision-language models on the Video-MME benchmark.** Eagle2.5 demonstrates consistent improvement as the number of input frames increases.

Eagle 2.5 is driven by both the advanced training strategy and data recipe. For training strategy, we introduce two core components for effective long-context learning: information-first sampling and progressive training.

- **Information-first sampling**. The information-first sampling strategy ensures the preservation of essential visual and semantic information through two mechanisms: (1) *Image Area Preservation*, which optimizes tiling to retain the majority of the original image area while maintaining aspect ratio fidelity, avoiding rigid aspect ratio constraints; and (2) *Automatic Degradation Sampling (ADS)*, which dynamically balances visual and textual inputs by prioritizing complete text retention while adaptively optimizing visual content to maximize context length utilization and preserve multimodal information.

- **Progressive training**. We employ a progressive mixed post-training approach, wherein context length is incrementally expanded during training, enhancing the model's ability to process inputs of varying sizes. This integrated strategy significantly improves information density over static sampling methods while ensuring consistent performance across diverse input types and lengths.

For data recipe, we embrace the "*diversity first, then quality*" principle in curating the training data pool. Our data recipe combines open-source data (including human-annotated data as well as synthetic video data) with our self-curated Eagle-Video-110K dataset, specifically designed to enhance long video understanding capabilities. We adopt a diversity-driven collection strategy, using multiple video sources and a similarity thresholding method to identify novel clips that maximize content diversity. Our dataset is distinguished by its dual annotation approach:

- A top-down story-level method that leverages human-annotated chapters as meaningful segments instead of traditional shot-level segmentation, producing dense captions that form the basis for comprehensive long-form QA pairs capturing the entire video's narrative structure;

- A complementary bottom-up clip-level approach that generates focused QA pairs for short clips using GPT-4o with diverse question types. To address the challenge of extending localized clip annotations to full-length videos, we implement anchors that incorporates temporal references and contextual elements without revealing answers, thereby letting models understand both overarching narratives and precise spatio-temporal details within videos.

## 2 Related Work

**Vision-language models.** Advancements in large language models (LLMs) [20, 1, 77] have significantly propelled visual understanding by integrating visual features, leading to the creation of Visual Language Models (VLMs) [52, 76, 58, 133]. Open-source VLMs [58, 60, 49, 97, 92, 18, 66, 50, 114, 67, 39] continue to achieve breakthroughs, often matching or exceeding the performance of state-of-the-art commercial models like GPT-4V/4o [77] and Gemini-1.5 [83]. The release of open-source VLMs [49, 95, 40, 38], complete with its training data and code base, has further

accelerated research in this area. However, most current VLMs primarily focus on short-context understanding, handling only a few images or short video clips at a time. Eagle 2.5 advances this field by concentrating on long-context visual understanding through a comprehensive exploration and development of training strategies and data recipes.

**Long-context VLMs.** Long-context VLMs were developed to address the challenges of processing large multimodal sequences. Currently, methods for long-context VLMs fall into two main categories. The first category involves specialized modules designed for context compression. Question-guided compressions [84, 57, 45, 98] or selection [117, 55, 116] methods extract question-related visual cues through an additional module, while various token reduction techniques [43, 117, 103, 53, 57, 124, 102, 62, 62, 61] aim to minimize the visual representation before LLM processing. The other category attempts to directly extend the context of LLMs. Works like LongVA [126], LongVILA [110], and LongViTA [85] extend the context length of LLMs to accommodate longer multimodal sequences. While promising, these approaches often underperform proprietary models, fail to show consistent performance improvements with increasing visual input, and have underexplored constraints on training strategies and data recipes. Our approach focuses on developing native long-context capabilities that enhance VLMs by exploring training data, formulations, and without introducing additional compression modules or suffering from performance inconsistencies observed in previous expansion attempts.

**Long-context multimodal data.** To enhance VLMs' long-context multimodal understanding capabilities, various datasets have been proposed. Some datasets focus on multimodal understanding of long documents [89, 96, 94, 80], such as slides and papers. However, they often lack temporal understanding. Other datasets [36, 105, 24, 82, 87, 88, 120, 121] emphasize the temporal coherence and information retrieval across long spans inherent in movies. Additionally, recent datasets [73, 13, 30, 129] covering domains further enhance VLMs' long-context multimodal understanding. Regarding the annotation methods for long-context multimodal datasets, early works [89, 96, 94, 36, 87, 73] relied on manual efforts. To reduce costs, some methods [80, 24, 82, 88, 120, 121, 30, 13, 129] use tools like GPT-4V [76] and Gemini [91] for automated or semi-automated annotation. Recent advancements in data construction emphasize hierarchical annotation strategies [30], which can preserve narrative structure in long videos. These advancements reflect a trend towards creating balanced datasets that effectively assess long-context multimodal understanding while managing creation costs.

# 3 Method

This section introduces the model architecture, training strategies, and data recipe of Eagle2.5.

## 3.1 Model Architecture

We design our proposed model as a versatile multimodal system capable of efficiently processing long-context information, rather than a specialized model solely optimized for handling extended multimodal inputs. To ensure adaptability and generalization across diverse tasks, we deliberately avoid incorporating tailored compression modules that might constrain the model's flexibility. Following the architecture of LLaVA [58], we employ an MLP projection layer to align vision embeddings from SigLIP [123] with the LLM representation space, as shown in

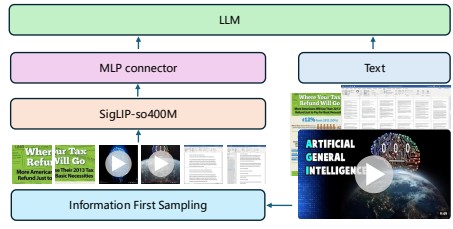

Figure 2: Tiling-based general multimodal system.

Fig. 2. In this work, we utilize the Qwen2.5 series models [93]. To effectively handle any-resolution images, we adopt the image tiling strategy, inspired by LLaVA-1.5 [59] and InternVL [16].

## 3.2 Training Strategy

Our approach contains two key components to achieve effective long-context training: first, an information-first sampling strategy that establishes optimal sampling criteria; and second, a progressive training schedule based on this strategy, which directs the entire model training process.

### 3.2.1 Information-First Sampling

In multimodal training, the sampling of visual content is essential. Multi-image documents typically comprise dozens of pages with ultra-high-resolution images, while video content can vary drastically in length - from mere seconds to hours. To effectively manage this diversity, we present **information-first sampling** to promote information preservation from both visual and semantic dimensions.

**Image area preservation (IAP).** Traditional tiling methods divide an image of size $W \times H$ into a rigid $r_w \times r_h$ grid of $s \times s$ tiles. While effective for handling high-resolution inputs, these approaches often distort the original image geometry through improper aspect ratio handling. For example, InternVL [16] imposes strict aspect ratio constraints that force image downsampling, undermining the purpose of tiling. To address this, we propose an area-prioritized tiling strategy that optimizes two key objectives:

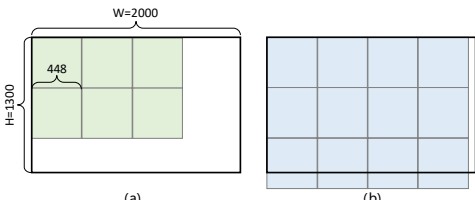

Figure 3: **Image area preservation.** Compared to the tiling strategy (a) from InternVL [92], our method (b) effectively retains a larger portion of the original image, especially for high-resolution inputs. This ensures that more comprehensive visual information is preserved, benefiting tasks that require fine-grained details.

- *Area Preservation*: Encourage maintaining at least 60% of the original area ($A_{\text{orig}} = WH$) in the tiled version ($A_{\text{new}} = r_w r_h s^2$).
- *Aspect Ratio Fidelity*: Align the tiling ratio $r_t = r_w/r_h$ with the original aspect ratio $r_{\text{orig}} = W/H$.

For candidate tiling ratios $\{(r_w, r_h) \mid r_w \times r_h \leq N\}$, we select the optimal configuration by:

$$\arg\max_{(r_w, r_h)} \left[ \underbrace{\min\left(\frac{A_{\text{new}}}{A_{\text{orig}}}, 0.6\right)}_{\text{Area penalty}} \cdot \underbrace{\min\left(\frac{r_t}{r_{\text{orig}}}, \frac{r_{\text{orig}}}{r_t}\right)}_{\text{Aspect ratio alignment}} \right] \tag{1}$$

This formulation imposes penalties when $A_{\text{new}} < 0.6 A_{\text{orig}}$ but avoids over-rewarding configurations where $A_{\text{new}} > 0.6 \times A_{\text{orig}}$. The aspect ratio term reaches maximum value 1 when $r_t = r_{\text{orig}}$, decaying symmetrically for deviations. A comparison between the strategies is shown in Fig. 3.

**Automatic degradation sampling.** VLMs require careful allocation of sequence length budgets between visual and textual inputs. Conventional **vision-context-centric** approaches sample visual content (e.g., video frames) at fixed rates or with predetermined counts, risking text truncation and suboptimal token allocation. We propose Automatic Degradation Sampling (**ADS**), an **all-context-centric** strategy that dynamically optimizes this balance.

Given a training sample $\mathcal{S} = \{S_{\text{visual}}, S_{\text{text}}\}$ with max sequence length $\mathcal{L}_{\text{max}}$, where $S_{\text{visual}}$ contains arbitrary combinations of images, videos, and multi-page documents: 1) Compute fixed text token length $\mathcal{L}_{\text{text}}$; and Derive fixed visual token budget: $\mathcal{L}_{\text{visual}} = \mathcal{L}_{\text{max}} - \mathcal{L}_{\text{text}}$. Thus, we keep the complete textual information by constricting the visual token budget.

For visual content optimization under $\mathcal{L}_{\text{visual}}$, we distinguish two types and optimize two key variables:

- **Images**: Optimize *maximal tile count per image t* to maximize spatial information of $M$ images.
- **Temporal content** (video/doc): Optimize *sampling count n* to maximize temporal coverage.

The constrained optimization problem is formulated as:

$$\max_{1 \leq t \leq 12,\ 1 \leq n \leq N_{\text{max}}} \sum_{i=1}^{M} L(t, I_i) + 256n$$
$$\text{s.t. } \sum_{i=1}^{M} L(t, I_i) + 256n \leq \mathcal{L}_{\text{vis}} \tag{2}$$

Where **optimization variables** are the tile count per image ($t$) and temporal sampling count ($n$), with **fixed parameters** including: total image instances $M$ (calculated from input), token function $L(t, I_i)$

used to calculate the tokens of $i$-th image $I_i$ under maximal tiling number $t$, predefined upper bounds $T_{\max} = 12$ (max tiles per image) and $N_{\max} = 2 \times$ duration$/1 \times$ pages (video/doc constraints). For temporal content, we do not use image tiling, thus the token quantity per temporal unit (frame/page) is $L(1, \cdot) = 256$.

Given that training samples typically exhibit mutually exclusive composition (predominantly images *or* temporal content), ADS employs a dual-phase degradation process to address the above optimization problem:

- *Temporal degradation*: Initially, we fix the max tile number $t = 1$ and focus on temporal sampling. We target a sampling rate of 2 FPS for videos, and the usage of all images for multi-image documents. We also require that each visual input has at least $N_{\min}$ frames; if this minimum cannot be met within the visual context budget, the sample is discarded. Formally, the maximally sampled temporal units is $n^* = \left\lfloor \frac{\mathcal{L}_{\text{visual}} - M \times 256}{256} \right\rfloor$.
- *Tiling degradation*: After deciding the number of frames, we dynamically adjust the tiling to maximize the use of available context. Let $\mathcal{T} = \{12, 8, 6, 4, 2, 1\}$ represent the possible tile configurations in decreasing order. We choose the highest tile configuration $t^*$ such that: $t^* = \max \left\{ t \in \mathcal{T} : \sum_{i=1}^{m} L(t, I_i) \leq (\mathcal{L}_V - n^* \times 256) \right\}$ This strategy preserves as much visual detail as possible while ensuring the full textual input is retained, thereby optimizing the overall learning signal.

This dual-phase approach guarantees complete text preservation while dynamically adapting visual resolution to available context budget, achieving superior information density compared to static sampling strategies.

### 3.2.2 Post-Training Schedule

We introduce a comprehensive post-training framework consisting of two complementary strategies. First, we establish a foundational mixed post-training approach, upon which we develop an enhanced progressive mixed post-training strategy to substantially improve model performance across varying context lengths.

- *Mixed post-training.* Since the model needs to efficiently process multimodal inputs of diverse lengths, maintaining consistent performance across variable context sizes is essential. Our ADS method adaptively adjusts each training sample to the maximum sequence length $\mathcal{L}_{\max}$, providing a frame-agnostic training paradigm. We implement a mixed training strategy with length-balanced packing [3] to optimize performance uniformly across the entire spectrum of context lengths.
- *Progressive mixed post-training.* For scenarios with large $\mathcal{L}_{\max}$ values, balancing the distribution of long and short sequences becomes computationally intensive, and achieving optimal performance through a single training iteration proves challenging. To address this limitation, we propose a progressive mixed training methodology that gradually exposes the model to increasingly larger $\mathcal{L}_{\max}$ values, systematically enhancing its capacity to process extended contexts. Compared to conventional mixed training, our method more effectively preserves the model's capabilities across different sequence lengths while safely generating diverse model variants at intermediate training stages. In our exeriment, we sequentially set $\mathcal{L}_{\max}$ to 32K, 64K and 128K.

### 3.3 Data Recipe

Our data recipe begins with open-source data. We embrace the "diversity first, then quality" principle and gather data from various open sources. This data mainly comprises high-definition multi-image/short videos, long videos, multi-page documents, and extensive text data. We also find that current open-source video data often lacks sufficient length. We thus propose a novel dataset, Eagle-Video-110K, to complement the length, as shown in Fig. 4.

### 3.3.1 Open-Source Long-Context Data

A model's capability is intrinsically linked to the diversity of its training data. Thus, gathering the most diverse data possible represents a core principle of this work, leading to two main strategies:

| Category | Dataset |
|---|---|
| Video Classification | Kinetics710 [9, 101], Something-Something-v2 [25], ActivityNet [8], HACS Segment [130], COIN [90], HIREST [122], FineAction [63], PortraitMode-400 [31] |
| Temporal Action Localization | ActivityNet [8], HACS Segment [130], FineAction [63], Ego4D-MQ [26], COIN [90], HIREST [122], Perception-Test [79] |
| Video Temporal Grounding | Charade-STA [23], QVHighlight [48], Ego4D-NLQ [26], Didemo [32], QueryD [75], MedVidQA [29], Youcook2 [132], FineVideo [22], ActivityNet [8], HACS Segment [130], FineAction [63], Ego4D-MQ [26], COIN [90], HIREST [122], Perception-Test [79], EgoExoLearn [37] |
| Dense Video Captioning | ActivityNet [8], Youcook2 [132], EgoExoLearn [37], ViTT [35], HIREST [122], COIN [90] |
| Temporal Segmentation | Breakfast [46], ViTT [35] |
| Temporal Reasoning | ActivityNet-RTL [34] |
| General Video QA | TVQA [47], CLEVRER [115], NextQA [108], SportsQA [51], LLaVA-Video [128], FineVideo [22], VideoGPT+ [69], Oops [21], Perception-Test [79], EgoTaskQA [42], CinePile [82], STAR [104] |
| Multi-Page Document | SlideVQA [89], DUDE [96], MP-DocVQA [94] |
| Video Captioning | ActivityNet [8], Youcook2 [132], Shot2story [30], Vript [112], LLaVA-Video [128], Momentos [99], FunQA [109], S-MiT [74], LLaVA-Hound [127], Ego4D-HCap [41], EgoExoLearn [37] |
| Long Text | LongAlign [6], LongReward [125] |

Table 1: Video, multi-page document, and long text dataset used in Eagle-2.5.

- *Human-annotated Data:* We integrate various open-source human-annotated datasets, including established video and image-document collections such as COIN [90] and SlideVQA [89], which can be directly considered as high-quality data.
- *Synthetic Video Data:* Considering that videos naturally contain long-context information, we incorporate open-source synthetic video data, such as LLaVA-Video [128]. These datasets are primarily annotated automatically using state-of-the-art models including GPT-4V/4o [76, 77], Claude-3 [2], and Gemini-1.5 Pro [83].

Combined with short-context data, all collected open-source datasets are summarized in Tab. 1. For convenience, we refer to this collective dataset as Open-Data.

### 3.3.2 Eagle-Video-110K

We curate Eagle-Video-110K to enhance long video understanding capabilities. Specifically, we first collect videos using a diversity-driven strategy. We then automatically annotate these videos using both *top-down* and *bottom-up* approaches to generate comprehensive story-level and fine-grained clip-level annotations, as shown in Fig. 5.

**Diversity-driven video collection.** We utilize several data sources for our video collection: Vidchapters [111], MiraData [44], InternVid-10M [100], Panda-70M [14], Vript [112], Shot2story [30], ViTT [35], and WebVid-10M [7], collectively referred to as $A$. Our approach prioritizes diversity, focusing on gathering a wide range of video content. For the current training dataset $B$, we use CLIP [81] to extract temporal features at a rate of 1 frame per second. Videos from both $A$ and $B$ are segmented into 10-second clips. We perform a pooling operation on each clip's frames to derive a representative feature vector. Let $\{b_i\}_{i=1}^{N_B}$ represent the clips from $B$,

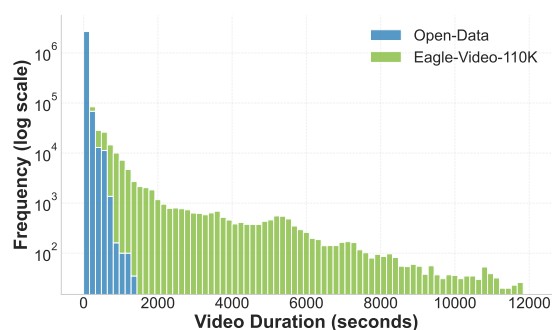

Figure 4: Comparison of video duration between open-source data and Eagle-Video-110K.

and $\{a_j\}_{j=1}^{N_A}$ represent those from $A$. We calculate the pairwise cosine similarity between clips from $B$ and $A$. For each clip $a_j$ in $A$, we identify its maximum similarity with any clip in $B$: $S_{\max}(a_j) = \max_{1 \leq i \leq N_B} S(b_i, a_j)$ We then introduce a similarity threshold $\tau = 0.5$. Clips in $A$ with $S_{\max}(a_j)$ below this threshold are considered most novel relative to $B$: $A_{\text{novel}} = \{a_j \in A \mid S_{\max}(a_j) < \tau\}$

The clips in $A_{\text{novel}}$ and their original videos are selected to enhance the diversity of our collection.

**Story-level video data.** We construct story-level annotations for long videos using a *top-down* approach. Unlike existing video datasets such as Shot2story [30], which employs shot detection to segment videos and construct storylines across shots, our methodology differs fundamentally. Shot-level segmentation often results in over-segmentation, producing excessively detailed annotations that are suboptimal for constructing coherent story-level text. Instead, we leverage human-annotated chapters as video segments, which provide more semantically meaningful annotations. We incorporate

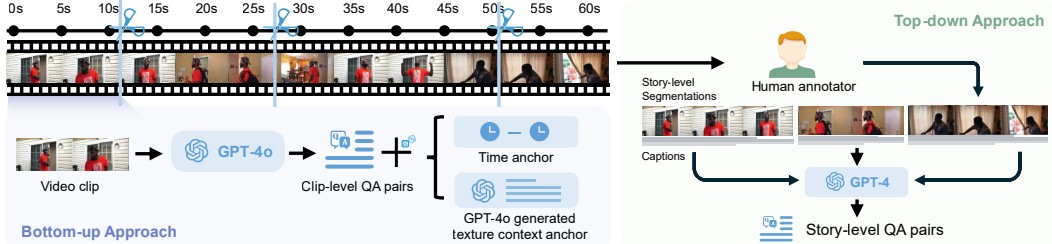

Figure 5: **Overview of our video annotation framework combining bottom-up clip-level and top-down story-level approaches.** The diagram illustrates our dual annotation strategy. In the bottom-up approach (left), short video clips are processed by GPT-4o to generate clip-level QA pairs enhanced with time anchors and textual context anchors. In the top-down approach (right), human annotators create story-level segmentations of longer videos, which are then captioned and processed by GPT-4 to generate comprehensive story-level QA pairs. This hierarchical methodology enables both fine-grained temporal understanding and high-level semantic comprehension of video content.

content from ViTT [35] and Vidchapters [111] among the selected videos and filter out any videos with fewer than two chapters to ensure they serve as effective story-level sources.

- *Chapter-level dense caption.* For a video divided into $N$ clips, where each clip spans from timestamp $a$ to $b$, we perform visual captioning for each segment individually. For each segment, frames are sampled at a rate of up to 2 frames per second, with a maximum of 50 frames. These sampled frames, together with user-provided segment titles, guide GPT-4o [77] in generating detailed visual descriptions focused on the content indicated by the titles.
- *Long-form QA generation.* Once visual descriptions for all segments are completed, we compile the captions for the entire video along with their corresponding time intervals and chapter titles. This aggregated information is provided to GPT-4 [1], which generates diverse question-answer pairs covering multiple question types.

**Clip-level video data.** Story-level video data typically emphasizes high-level semantic information that unfolds over extended periods. However, for general queries, it is often necessary to focus on localized spatiotemporal details. To address this need, we propose a *bottom-up*, computationally efficient automatic annotation method. This approach enables the generation of short clip annotations and facilitates the conversion of segment-level annotations into video-level ones by incorporating temporal and contextual anchors.

- *Clip-level video QA generation.* We generate QA pairs for each short clip in dataset $A$ based on various question types. Specifically, we sample frames from each short clip at a rate of up to 2 frames per second and input them into GPT-4o. From a predefined question type pool, we randomly select five question types and prompt the model to generate corresponding question-answer pairs.
- *Clip-to-video QA conversion.* Since annotations for individual clips are designed for localized queries, conflicts may arise when these queries are extended to the entire video. To address this issue, we introduce two types of anchors for each clip-question pair: (1) We directly incorporate time intervals into questions to establish temporal references; (2) Using GPT-4o, we generate textual context anchors that provide additional information without revealing the answers.

## 4 Experiments

### 4.1 Comparison with State-of-the-Art VLMs

**Video benchmarks.** As shown in Tab. 2, Eagle2.5-8B demonstrates strong performance across multiple video understanding benchmarks. It achieves 74.8 on MVBench [54], 82.0 on Perception_test [78] and 72.2 on EgoSchema, outperforming similar-sized models like InternVL2.5-8B [92] (72.0, -, -) and Qwen2.5-VL-8B [97] (69.6, 70.5, 65.0). Eagle2.5-8B excels in MLVU [131] (77.6) and LongVideobench [106] (66.4), surpassing even InternVL2.5-78B (75.7, 63.6). For VideoMME (w/o subtitle), the performance of Eagle 2.5 (72.4) significantly surpasses models of the same size and is extremely close to the 72B parameter model. On CG-Bench [11], it scores 55.8, 46.6, 45.6, 13.4 across metrics, exceeding Claude-3.5-Sonnet [2] (56.5, 40.3, 35.6, 4.17) and Gemini-1.5-Pro [83]

| Model | MVBench | Perception_test | EgoSchema | MMB-Video | MLVU | LVBench | Video-MME | | CG-Bench | | | | HourVideo | | Charade-STA |
|---|---|---|---|---|---|---|---|---|---|---|---|---|---|---|---|
| | - | Val | fullset | - | Val | Val | w/o subtitle | w subtitle | Clue | Long | Open | mIoU | Dev | Test | mIoU |
| *Closed-Source Models* | | | | | | | | | | | | | | | |
| GPT-4o-0806 [77] | - | - | - | 1.63 | - | 66.7 | 71.9 | 77.2 | 58.6 | 44.9 | 39.2 | 5.73 | - | - | 35.7 |
| Claude-3.5-Sonnet [2] | - | - | - | - | - | - | 60.0 | 62.9 | 56.5 | 40.3 | 35.6 | 4.17 | - | - | - |
| Gemini-1.5-Pro [83] | - | - | 72.2 | 1.30 | - | 64.0 | 75.0 | 81.3 | 50.9 | 37.8 | 28.7 | 3.85 | 37.2 | 37.4 | - |
| Gemini-2.5-Pro [19] | - | - | - | - | 81.2 | 69.2 | 87.0 | - | - | - | - | - | - | - | - |
| Seed1.5-VL [28] | 74.3 | - | - | - | 81.8 | 64.6 | 77.6 | - | - | - | - | - | - | - | 64.7 |
| *Publicly Available Models* | | | | | | | | | | | | | | | |
| MiniCPM-V2.6-8B [113] | - | - | - | - | - | - | 60.9 | 63.7 | 44.4 | 29.9 | 26.3 | 2.27 | - | - | - |
| LongVILA-8B [15] | 67.1 | 58.1 | 67.7 | - | - | 57.1 | 60.1 | 65.1 | 47.5 | 34.3 | 26.6 | - | - | - | - |
| InternVL2.5-8B [17] | 72.0 | - | - | 1.68 | 68.9 | 60.0 | 64.2 | 66.9 | - | - | - | - | - | - | - |
| LLaVA-Video-8B [129] | 58.6 | 67.9 | 57.3 | - | 70.8 | 58.2 | 63.3 | 69.7 | - | - | - | - | - | - | - |
| Qwen2.5-VL-8B [5] | 69.6 | 70.5 | 65.0 | 1.79 | 70.2 | 56.0 | 65.1 | 71.6 | 44.5 | 35.5 | 24.1 | 2.48 | - | - | 43.6 |
| VideoChat-Flash-8B [56] | 74.0 | 76.2 | - | - | 74.6 | 64.7 | 65.3 | 69.7 | 52.8 | 43.1 | 37.5 | 1.49 | - | - | - |
| InternVL2.5-78B [17] | 76.4 | - | - | 1.97 | 75.7 | 63.6 | 72.1 | 74.0 | 59.5 | 44.2 | 34.2 | 3.90 | - | - | - |
| Qwen2.5-VL-72B [5] | 70.4 | 73.2 | 76.2 | 2.02 | 74.6 | 60.7 | 73.3 | 79.1 | - | - | - | - | - | - | 50.9 |
| LLaVA-Video-72B [129] | 64.1 | 74.3 | 65.6 | - | 74.4 | 61.9 | 70.6 | 76.9 | - | - | - | - | - | - | - |
| Eagle2.5-8B | 74.8 | 82.0 | 72.2 | 1.94 | 77.6 | 66.4 | 72.4 | 75.7 | 55.8 | 46.6 | 45.6 | 13.4 | 44.5 | 41.8 | 65.9 |

Table 2: **Comparison with SoTA models on Various Video Benchmarks**. We sample each video at 2 FPS by default and disable tiling, and limit the minimum sampling frame number to 8 frames. Among them, the maximum frame number of Video-MME is 512, and the others are 256. Perception-Test turns on tiling to enable high-resolution testing.

| Model | DocVQA | ChartQA | InfoVQA | TextVQA | OCRBench | MMstar | RWQA | AI2D | MMMU | MMB$_{1.1}$ | MMVet | HallB | MathVista | **Avg** |
|---|---|---|---|---|---|---|---|---|---|---|---|---|---|---|
| | Test | Test | Test | Val | Test | Test | Test | Test | Val | Test | Test | Test | Test-Mini | **Score** |
| *Closed-Source Models* | | | | | | | | | | | | | | |
| GPT-4o-0806 [77] | 92.8 | 85.7 | 79.2 | 77.4 | 736 | 64.7 | 75.4 | 84.6 | 69.1 | 83.1 | 69.1 | 55.0 | 63.8 | 74.9 |
| Claude-3.5-Sonnet [2] | 95.2 | 90.8 | 74.3 | 74.1 | 788 | 65.1 | 60.1 | 81.2 | 68.3 | 80.9 | 70.1 | 55.5 | 67.7 | 74.0 |
| Gemini-1.5-Pro [83] | 93.1 | 87.2 | 81.0 | 78.8 | 754 | 59.1 | 67.5 | 79.1 | 62.2 | 74.6 | 64.0 | 45.6 | 63.9 | 71.7 |
| *Publicly Available Models* | | | | | | | | | | | | | | |
| MiniCPM-V2.6-8B [113] | 90.8 | 82.4 | - | 80.1 | 852 | 57.5 | 65.0 | 82.1 | 49.8 | 78.0 | 60.0 | 48.1 | 60.6 | - |
| LLaVA-One-Vision-8B [49] | 87.5 | 80.0 | 68.8 | - | 622 | 61.7 | 66.3 | 81.4 | 48.8 | 80.9 | 57.5 | 31.6 | 63.2 | - |
| InternVL2.5-8B [17] | 93.0 | 84.8 | 77.6 | 79.1 | 822 | 62.8 | 70.1 | 84.5 | 56.0 | 83.2 | 62.8 | 50.1 | 64.4 | 73.1 |
| Qwen2.5-VL-8B [5] | 95.7 | 87.3 | 82.6 | 84.9 | 864 | 63.9 | 68.5 | 83.9 | 58.6 | 82.6 | 67.1 | 52.9 | 68.2 | 75.6 |
| LLaVA-One-Vision-72B [49] | 91.7 | 83.7 | 74.9 | - | 741 | 66.1 | 71.9 | 85.6 | 56.6 | 84.5 | 60.6 | 47.5 | 68.4 | - |
| LLaMa-3.2-90B-Vision [20] | 90.1 | 85.5 | - | - | 783 | 55.3 | - | - | 60.3 | 77.3 | 64.1 | 44.1 | 57.3 | - |
| Eagle2.5-8B | 94.1 | 87.5 | 80.4 | 83.7 | 869 | 66.2 | 76.7 | 84.5 | 55.8 | 81.7 | 62.9 | 54.7 | 67.8 | 75.6 |

Table 3: **Comparison with SoTA models on Various Image Benchmarks.** The avg score is computed as the average of all benchmark scores, with OCRBench score divided by 10.

(50.9, 37.8, 28.7, 3.85). With 44.5 on HourVideo [10] dev set and 41.8 on test set, all surpassing Gemini-1.5-Pro [83]. Finally, on Charade-STA [23], Eagle 2.5 outperforms other models significantly, demonstrating strong temporal perception capabilities. Eagle2.5-8B shows effective long-form video understanding, highlighting its robust visual reasoning using less parameters.

**Image benchmarks.** As shown in Tab. 3, Eagle2.5-8B demonstrates competitive performance across diverse image understanding benchmarks. It achieves strong results on document understanding (94.1 on DocVQA [71]), chart interpretation (87.5 on ChartQA [70]), and general information extraction (80.4 on InfoVQA [72], 83.7 on TextVQA [86]). The model also performs well in optical character recognition with 869 on OCRBench [65], comparable to other models in its category. Eagle2.5-8B shows balanced capabilities across multimodal general perception and reasoning tasks, scoring 66.2 on MMstar [12], 76.7 on RWQA [107], and 81.7 on MMB$_{1.1}$ [64], and 62.9 on MMVet [118]. Its performance extends to knowledge domain (55.8 on MMMU [119], 84.5 on AI2D [33]), visual hallucination benchmark (54.7 on HallB [27]), and mathematical reasoning (67.8 on MathVista [68]). Overall, Eagle2.5-8B achieves a competitive 75.6 average score, demonstrating its effectiveness as a versatile vision-language model that balances performance across various visual understanding tasks.

## 4.2 Ablation Studies

In this section, we conduct experiments on various benchmarks to evaluate our method. We mainly design experiments to study the following questions.

**Q1: How do video and image data influence each other's benchmarks?** Tab. 4 studies the impact of long context data on the image benchmark performance. We compare the image benchmark performance without training with long-context data and with training long-context data under different $\mathcal{L}_{\max}$. The results show that increasing the long-context data, under our training recipe, does not harm the short-context images and even slightly benefits it. To assess the impact of image data and pre-training on video benchmarks, we conduct a comparison using the $\mathcal{L}_{\max} = 32K$ model.

| Training & Data recipe | DocQA | ChartQA | InfoVQA | TextVQA | OCRBench | MMstar | RWQA | AI2D | MMMU | MMB$_{1.1}$ | MMVet | HallB | MathVista | Avg |
| --- | --- | --- | --- | --- | --- | --- | --- | --- | --- | --- | --- | --- | --- | --- |
| | Val | Test | Val | Val | Val | Test | Test | Test | Val | EN-Val | Test | Test | Test-Mini | Score |
| Eagle2.5-S2 | 92.6 | 88.3 | 78.8 | 84.6 | 868 | 66.5 | 74.4 | 85.5 | 54.0 | 85.5 | 57.3 | 53.4 | 65.1 | 74.8 |
| Eagle2.5-S2+Eagle2.5-S2, $\mathcal{L}_{\max} = 32K$ | 92.3 | 86.6 | 77.6 | 82.8 | 861 | 66.7 | 75.9 | 83.7 | 55.5 | 84.8 | 63.6 | 55.4 | 68.3 | 75.3 |
| Eagle2.5-S2+Eagle2.5-S2, $\mathcal{L}_{\max} = 64K$ | 92.5 | 87.0 | 78.4 | 83.9 | 865 | 66.8 | 76.8 | 83.9 | 55.7 | 85.2 | 63.3 | 55.2 | 67.3 | 75.6 |
| Eagle2.5-S2+Eagle2.5-S2, $\mathcal{L}_{\max} = 128K$ | 93.2 | 87.5 | 78.5 | 83.7 | 869 | 66.2 | 76.7 | 84.5 | 55.8 | 85.5 | 62.9 | 54.7 | 67.8 | 75.7 |

Table 4: Impact of long-context data on performance of image benchmarks.

| Training & Data recipe | MVBench | MLVU | Video-MME |
| --- | --- | --- | --- |
| | - | Val | w/o subtitle |
| S1→S2 | 70.4 | 67.4 | 64.9 |
| S1→S1.5→S2 (OD+EV-110K) | 72.9 | 70.9 | 65.2 |
| S1→S1.5→S2 (Image+OD+EV-110K) | 73.1 | 71.5 | 65.4 |

| Recipe | InfoVQA | DocQA | TextVQA | PT | MLVU | Video-MME |
| --- | --- | --- | --- | --- | --- | --- |
| | Val | Val | Val | Val | Val | w/o subtitle |
| baseline | 77.6 | 92.3 | 82.8 | 76.3 | 71.5 | 65.4 |
| w/o IAP | 76.2 | 91.9 | 82.4 | 73.3 | 71.2 | 64.9 |
| w/o ADS | 77.0 | 92.1 | 82.8 | 75.5 | 70.1 | 65.0 |

Table 5: The impact of image data and pretraining on the performance of video benchmarks. S1/S1.5 denotes the stage-1 and stage-1.5 similar to Eagle2 [3]. "OD" is short for Open-Data.

Table 6: The impact of information-first sampling on performance of image and video benchmarks. The baseline is equipped with IAP and ADS strategy. "PT" is short for PereptionTest.

For each benchmark, we sampled at 2FPS, ensuring a maximum of 32 frames. As shown in Tab. 5, extensive image pre-training significantly enhances performance on short video benchmarks like MVBench, as well as on the relatively simple long video benchmark, MLVU. However, for the more challenging and held-out long video benchmark, Video-MME, the improvements are less pronounced.

**Q2: The effect of information-first sampling on performance?** Tab. 6 illustrates the impact of the information-first sampling strategy on image and video tasks. Without the Image Area Preservation strategy, high-resolution image benchmarks like InfoVQA and fine-grained video benchmarks such as Perception-test suffer significant performance degradation. The effect on other benchmarks is less pronounced. While the Automatic Degradation Sampling strategy offers convenience for processing various visual inputs, experiments indicate that omitting it poses a risk. The vision-context-centric strategy may truncate supervision signals, leading to performance loss.

**Q3: The impact of different post-training schedules?** Tab. 7 illustrates the performance impact of progressive mixed training from 32K to 64K compared to direct 64K mixed training on the video benchmarks. The results demonstrate that progressive training outperforms direct 64K mixed training, possibly due to two reasons: 1) Direct 64K hybrid training disperses samples across the 64K space, diluting the focus on shorter contexts.

| Training & Data recipe | MVBench | MLVU | Video-MME |
| --- | --- | --- | --- |
| | - | Val | w/o sbutitle |
| 32K→64K, Open-Data | 73.0 | 74.5 | 68.1 |
| 64K, Open-Data | 71.3 | 74.0 | 67.9 |
| 32K→64K, Open-Data+EV-110K | 73.9 | 75.1 | 68.8 |

Table 7: The impact of Eagle-Video-110K dataset and different post-training schedules on the performance of video benchmarks.

2) Some longer samples are challenging to learn without a gradual learning process that transitions from easy to difficult. Fig. 6 shows the effect of progressive mixed training on the Video-MME benchmark. It reveals that as progressive training advances, the model's capacity to process more frames is gradually enhanced.

**Q4: The impact of Eagle-Video-110K data on performance?** We assess the impact of Eagle-Video-110K on model performance. Table 7 shows that it improves several long and short video benchmarks. Figure 6 demonstrates that training with Eagle-Video-110K enhances the model's ability to process many frames ($\geq 128$ frames) by incorporating long videos absent from the open-source training dataset.

## 5 Conclusion

In this work, we present Eagle 2.5, an advanced vision-language model family designed for long-context multimodal understanding. Through innovative training approaches - including information-first sampling and progressive mixed post-training - combined with our dual-annotated Eagle-Video-110K dataset, we significantly enhance long-context

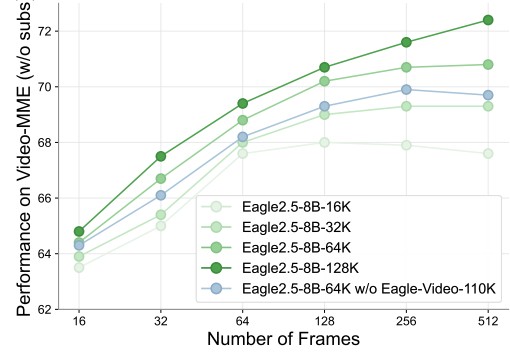

Figure 6: The impact of Eagle-Video-110K dataset and different post-training schedules on the performance of Video-MME.

comprehension capabilities. Eagle 2.5 achieves leading performance on video and high-resolution image benchmarks, matching larger models like GPT-4V and Gemini 1.5 Pro despite its smaller size. With advanced training strategies and diverse data, Eagle 2.5 sets a strong foundation for future research, paving the way for efficient and versatile VLMs in complex real-world scenarios.

**Limitations.** The training of Eagle2.5 required substantial computational resources, specifically a cluster of 128 H100 GPUs. This high resource demand may limit the reproducibility and accessibility of our approach for researchers or practitioners without access to large-scale GPU infrastructure. Future work could explore more resource-efficient training strategies or model architectures to reduce computational requirements.

## 6 Acknowledgement

This work is supported by the National Key R&D Program of China (No. 2022ZD0160900), the Basic Research Program of Jiangsu (No. BK20250009), the National Natural Science Foundation of China (Grant No.62372223 and U24A20330), and in part by Nanjing University-China Mobile Communications Group Co., Ltd. Joint Institute under Grant NJ20250037.

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
