# OpenReview forum: "Eagle 2.5: Boosting Long-Context Post-Training for Frontier Vision-Language Models"
_NeurIPS.cc/2025/Conference — NeurIPS 2025 poster_

### Official Review · Reviewer_bMM9 · 2025-06-29

**Clarity:** 3
**Significance:** 3
**Originality:** 3
**Rating:** 4
**Confidence:** 4

**Summary:**

This paper proposes a frontier multi-modal large language model named Eagle2.5, which focuses on long-context multi-modal learning. Eagle2.5 is a generalist framework to addresses the challenges in long video comprehension and high-resolution image understanding. It proposes Automatic Degrade Sampling and Image Area Preservation to preserve contextual integrity and visual details. Also, the work also proposes a diverse and high-quality dataset named Eagle-Video-110K to facilitating long-context training. Extensive experiments demonstrate the effectiveness of Eagle2.5.

**Questions:**

In addition to the questions in Weakness part, I have another important question.

In Table 3, Eagle2.5 does not perform better than Qwen2.5-VL-8B in terms of the overall average metric. For example, Eagle2.5 is inferior to Qwen2.5-VL-8B on DocVQA. Could you please show me some insights or conjectures about the reasons of these phenomena? I think some explanation or analysis will surely improve the quality of the paper.

**Ethical Concerns:**

["NO or VERY MINOR ethics concerns only"]

**Final Justification:**

I initially gave a positive rating. After rebuttal, my concerns have been addressed. Thus I recommend accepting this paper.

**Limitations:**

yes

**Quality:**

3

**Strengths And Weaknesses:**

The paper proposes a new frontier multi-modal large language model to address long-context modeling, and their experiments have demonstrated the effectiveness of the proposed model. Thus, I think this paper does make some contributions.

However, I have some concerns about the techniques and experiments, which are listed as follows:
- In my understanding, the proposed Image Area Preservation involves more patches/tokens from the the original videos/images, such that it would cover more complete content in videos/images. However, it will surely involve more computation cost. Have you analyze the budget?
- Eagle-Video-110K involves synthetic video data, whose captions are automatically generated by advanced MLLMs like GPT-4o and Gemini-1.5-Pro. Have you make some efforts to ensure the quality of generated captions as these MLLMs have hallucinations.
- In Line 245, why you use CLIP? Could you please give me some reasons? Have you try any other encoders?
- I think it will be better that you can highlight the best/second-best results in experiment tables, e.g., Table 2 and 3.

---

> ### Author Rebuttal · Authors · 2025-07-31
>
> ## About the computation cost of IAP sampling
>
> **Computation cost does not necessarily increase**. Our IAP, which maintains the same performance as traditional tiling for regular images, does not increase the actual number of visual tokens. IAP primarily targets long-context sampling of images, particularly for images with long-tail aspect ratios similar to those found in InfoVQA. For example, for an image with a height and width of (1366, 6350) in InfoVQA, the number of visual tokens generated by Eagle2.5, InternVL3, and Qwen2.5VL (these models use 28x28 compression ratio for each image) is as follows.
>
> |  | vision token (1366, 6350) | inference speed (1366, 6350) | InfoVQA val set |
> | --- | --- | --- | --- |
> | InternVL-3 8B | 4864 | 0.45s | 76.8 |
> | Qwen2.5VL 8B | 10848 | 0.78s | 82.6 |
> | Eagle-2.5 8B | 5376 | 0.47s | 80.4 |
>
> Compared to traditional tiling methods like InternVL3 and LLaVA, IAP preserves more contextual information for large images, resulting in better results (as shown in Table 6 in the main paper), while only slightly increasing computational overhead (0.45s → 0.47s). Compared to Qwen2.5VL's anyres sampling strategy, Eagle2.5-8B saves about 50% vision tokens, yet its performance is only slightly lower (qwen2.5vl: 82.6).
>
> ## About the data quality
>
> Multiple works[1][2] have proved that integrating or collaborating multiple VLM models can reduce the response illusion and accuracy of queries. Hence, to curb hallucination in automatically generated captions and QA pairs, we performed a mutual review cycle in which GPT-4o and Gemini-1.5-Pro iteratively scored and revised each other’s annotations until consensus was reached. This procedure markedly improved caption fidelity, but it also introduced substantial **time and money costs**. In pilot studies with a 7B-parameter LLM, the marginal gain in downstream performance did not justify the extra cost. Consequently, we prioritized incorporating a broader range of newly rollout of GPT4o/Gemini 1.5 Pro—after basic sanity checks—into the full training set, deferring more intensive refinement to future work.
>
> ## Why use CLIP for video clip retrieval?
>
> We initially considered several models as potential options, including CLIP, SSCD, InternVideo, and SigLIP. However, after careful evaluation, we ultimately selected CLIP for our deployment. Our decision was based on several factors. First, although SSCD is effective for capturing detailed visual textures, we observed that the queries in existing video benchmarks are more semantic in nature, rather than focusing on fine-grained visual details. InternVideo and SigLIP are also strong candidates, but they come with higher computational overhead, which requires further consideration in our deployment context. In contrast, CLIP offers a good balance between semantic understanding and computational efficiency, and its ONNX format allows for fast and easy inference.
> In summary, while we considered multiple alternatives, we chose CLIP because it best meets the semantic requirements of our tasks while ensuring efficient inference.
>
> ## About result highlighting in table 2 and 3
>
> We appreciate the reviewer’s suggestion regarding the presentation of experimental results. We will revise the tables in the final version to highlight the best and second-best performance values for better clarity.
>
> ## About the performance comparison with Qwen 2.5 VL
>
> Qwen2.5 VL differs from Eagle 2.5 in many aspects. It uses a larger vision encoder than Eagle 2.5 (Qwen vision encoder 600m vs. siglip-400m used in Eagle2.5) for visual encoding. In addition, and most importantly, it samples much more pre-training data than Eagle 2.5 and does not publish the training data list. Qwen2.5VL trains on 18T vision-language tokens. Eagle 2.5 trains on a total of 0.67T vision-language tokens across its four stages. Consequently, the gap we observe on DocVQA and the overall average metric is likely driven less by architectural shortcomings in Eagle 2.5 but more by this stark disparity in encoder capacity and pre-training scale; we expect that scaling Eagle 2.5 to comparable data volumes and a higher-capacity vision backbone would substantially narrow the performance difference.
>
> [1] Enhancing LLMs with an Ensemble of Critics for Mitigating Hallucination
>
> [2] LVAgent: Long Video Understanding by Multi‑Round Dynamical Collaboration of MLLM Agents

---

> > ### Comment · Reviewer_bMM9 · 2025-08-03
> >
> > Thanks for the authors' feedbacks. My concerns have been addressed. I will maintain my positive rating.

---

### Official Review · Reviewer_YT9W · 2025-07-02

**Clarity:** 3
**Significance:** 3
**Originality:** 3
**Rating:** 5
**Confidence:** 4

**Summary:**

Eagle2.5, a Vision-Language Model (VLM) aimed at advancing long-context multimodal understanding. The authors tackle the challenges of processing long videos and high-resolution images by introducing a novel training framework and dataset. The core contributions are threefold: An "Information-First Sampling" strategy to dynamically balance and preserve visual and textual data, a "Progressive Mixed Post-Training" schedule that incrementally increases the model's context length , the Eagle-Video-110K dataset, which provides much-needed long-form video content with a dual-annotation approach for both narrative and detail. Through extensive experiments, the 8B-parameter Eagle2.5 is shown to achieve state-of-the-art performance, often matching or outperforming significantly larger models.

**Questions:**

1. Regarding the Nature of IAP/ADS:The IAP and ADS methods are query-agnostic, pre-processing heuristics, applied before the model sees the user's actual question. Given that the optimal strategy for information retention is highly context-dependent , how does this approach mitigate the risk of discarding crucial visual details before their relevance can be assessed in light of the specific question being asked? How to prove this heuristic actually preserve more information than direct sample and tiling method? Is it necessary to utilize this sampling strategy during training for it to be effective at inference, or could it be applied post-hoc to other pre-trained VLMs?
2. Regarding Analysis of IAP/ADS: To better understand the behavior of the sampling strategies, would it be possible to provide a specific case study? For instance: A visualization of a high-resolution image tiled using IAP versus a baseline method. A quantitative example of the trade-off made by ADS for a specific long-video sample that contains lengthy text, showing how many frames are preserved versus discarded.
3. Is there a clear plan to open-source the methods and dataset? I believe this could greatly benefit the field.

**Ethical Concerns:**

["NO or VERY MINOR ethics concerns only"]

**Final Justification:**

After carefully reading the authors' response, the other reviews, and the ensuing discussion, my concerns have been addressed, I believe the work is solid and meets the bar for acceptance. Therefore, I have raised my score to 5 (Weak Accept).

**Limitations:**

yes

**Quality:**

3

**Strengths And Weaknesses:**

**Strengths:**
1. Significance: The paper tackles a highly relevant and challenging frontier in multimodal AI: long-context visual understanding.
2. Noval Training Strategy and Data Curation: The paper introduces a well-motivated progressive training schedule to effectively scale context length. Furthermore, the development of the Eagle-Video-110K dataset addresses a recognized scarcity of long-form video data for training, and its dual annotation strategy is a thoughtful approach to capturing both semantic and temporal details.
3. Rigorous Evaluation: The claims are substantiated by extensive experiments across a wide array of benchmarks and detailed ablation studies that validate the effectiveness of each component of the proposed framework.
4. Competitive Performance: The 8B model demonstrates strong empirical results, achieving performance on par with or exceeding much larger open-source and proprietary models.
5. Clarity and Organization: The paper is well-written, logically structured, and easy to follow. The authors clearly articulate the problems with existing long-context VLMs and present their solutions in a coherent manner.

**Weaknesses:**
1. Lack of Qualitative Analysis: The paper would be strengthened by a qualitative analysis of its core sampling strategies. It currently lacks case studies or visualizations to provide intuition for how IAP and ADS make decisions. For example, showing how an image is tiled by IAP or which frames are dropped by ADS in a given scenario would offer valuable insight into their practical behavior.
2. The dataset proposed in the article holds significant importance, as it appears to contribute most noticeably to the improvement in results. However, the manuscript does not seem to provide a clear plan for the dataset's release, which could impact the reproducibility and overall significance of the work.

---

> ### Author Rebuttal · Authors · 2025-07-31
>
> ## Risk of Dropping Task-Critical Visual Cues
>
> Before being fed into the LLM, all visual operations—such as sampling, slicing, visual encoding, visual token compression, and the visual-to-LLM projection—are query-agnostic. Consequently, two key factors determine the effectiveness of visual cue extraction: (1) preserving the integrity of the original input source, which directly sets the upper bound of visual feature information available to the LLM; and (2) providing complete textual semantic supervision, ensuring the visual encoder can extract universal visual features adaptable to any query. Our IAP ensures the upper limit of visual information, while ADS complements it by guaranteeing textual semantic supervision. Through end-to-end training, the visual encoder becomes task-agnostic and capable of extracting general visual features for query-aware decoding in the LLM.
>
> ## How to prove heuristic method actually preserve more information.
>
> We show this through practical experimental results. For IAP, we scan all images in the training set, compare the results of IAP and traditional tiling methods from InternVL and LLaVA, and remove results that did not match the tiling. We calculate the average coverage of the original image after tiling, as shown in the table below.
>
> |  | IAP | vanilla tiling |
> | --- | --- | --- |
> | Average Pixel Coverage | 73.82% | 44.19% |
>
> The results show that IAP can retain more pixels input to the model.
>
> For ADS strategy, we scan all video data in the training data, compare the results of ADS and non-ADS sampling strategies. non-ADS strategy samples frames at fixed FPS and is limited by the maximum number of frames. We calculate the label truncation rate and sample discard rate, as shown in the table below. We set `max_len = 16k`.  ADS minimizes both the label truncation rate and the sample dropout rate by adaptively calculating the number of sampling frames.
>
> |  | Average label truncation rate | Average sample discard rate |
> | --- | --- | --- |
> | ADS | 0.2% | 0.4% |
> | non-ADS (fps2, max 8 frames) | 0.5% | 0.1% |
> | non-ADS (fps2, max 16 frames) | 2.1% | 0.3% |
> | non-ADS (fps2, max 32 frames) | 11.3% | 4.8% |
> | non-ADS (fps2, max 48 frames) | 22.3% | 10.4% |
>
> In this table, the label truncation rate counts samples with partially truncated labels, and the sample discard rate counts samples with completely truncated labels.
>
> ## Need for sampling strategy at training and testing time
>
> IAP/ADS can be applied post-hoc to any pre-trained VLM because it is a pure input-layer transformation; no model parameters are modified. Nevertheless, matching the sampling distribution during training yields the best performance. We therefore recommend "train-time + test-time" for new models. ADS ensures that, given any input of multiple rounds of text-and-image dialogues, the processor automatically performs heuristic sampling of visual tokens according to the set `max_len`, thereby ensuring that labels are not truncated and reducing the cost of hyper-parameter tuning.
>
> ## Visualization of IAP
>
> As is shown in the Figure 3 of the paper, when the input image size is 1300×2000 and the maximum number of tiles 12, the target aspect ratios include:[(1, 1), (1, 2), (2, 1), (3, 1), (1, 3), (2, 2), (4, 1), (1, 4), (5, 1), (1, 5), (1, 6), (6, 1), (3, 2), (2, 3), (7, 1), (1, 7), (4, 2), (2, 4), (1, 8), (8, 1), (1, 9), (3, 3), (9, 1), (2, 5), (5, 2), (10, 1), (1, 10), (11, 1), (1, 11), (12, 1), (3, 4), (4, 3), (1, 12), (6, 2), (2, 6)].
>
> In InternVL's tiling method, the closest ratio to the image aspect ratio of 13:20 is 2:3, and thus the final tiles cover an area of 896×1344 pixels.
>
> In contrast, IAP (our method) considers not only aspect ratio proximity but also the matching degree of tile area to the original image, which helps avoid generating excessive redundant tiles or overly small tiles that lead to information loss. According to Eq. (1), compared to a score of 0.4516 for the 2:3 configuration, the 3:4 configuration achieves a higher score of 0.5200. As a result, our method selects the 3:4 layout and tiles an area of 1344 × 1792, which better preserves visual information and reduces unnecessary padding or information loss.
>
> ## **About release plan**
>
> The model weight and inference code are open to the public now. The training code and dataset are under legal review. They will be released when the legal review is done.

---

> > ### Comment · Reviewer_YT9W · 2025-08-03
> > **Response to the Author’s Rebuttal**
> >
> > Thank you to the authors for the detailed response, which has addressed my concerns.

---

### Official Review · Reviewer_4qXs · 2025-07-03

**Clarity:** 3
**Significance:** 2
**Originality:** 2
**Rating:** 4
**Confidence:** 4

**Summary:**

The paper introduces Eagle 2.5, an 8 B-parameter vision-language model (VLM) aimed at native long-context reasoning across both video and high-resolution image domains. Key ingredients are:

1) **Information-First Sampling (IFS)** – consisting of Image-Area Preservation (IAP) to keep ≥ 60 % of the original pixels during tiling, and Automatic Degradation Sampling (ADS) that dynamically trades visual tokens for text tokens so the text prompt is never truncated.

2) **Progressive training** – the context window of the model is gradually expanded, which the authors show yields better length-robustness than a single long-window run.

3) **Eagle-Video-110 K**, a 110 k-clip dataset with dual story-level and clip-level annotations to enhance long-form video understanding capabilities.

**Questions:**

Please see Strengths and Weaknesses Section for a list of questions.

**Ethical Concerns:**

["NO or VERY MINOR ethics concerns only"]

**Final Justification:**

Thank you authors for the rebuttal. Authors have addressed my concerns.

I will maintain my positive rating.

**Limitations:**

Yes.

**Quality:**

3

**Strengths And Weaknesses:**

Strengths:
1) **Timely focus on long-form video understanding.** Scaling vision-language models to hour-long content is a pressing challenge; meaningful progress here benefits both research and downstream applications.

2) **Valuable resource.** The Eagle-Video-110 K dataset (≈110 k clips with story- and clip-level annotations) should become a useful test-bed for future work on extended-duration reasoning and retrieval.

3) **Impressive efficiency/accuracy frontier (Table 2).** The proposed 8B parameter model matches or beats far larger models on Video-MME, LVBench, and, notably, outperforms Gemini 1.5 on the challenging HourVideo benchmark.


Weaknesses:
1) **HourVideo breakdown:** Could you provide per-category results for HourVideo (analogous to Table 2 in the HourVideo [A] paper)? This would clarify whether gains come primarily from visual reasoning, or other tasks such as summarization or perception.

2) **Reproducibility & release plan:** Will you release all training code, model weights, and the full Eagle-Video-110 K dataset to the community?

3) **Terminology clarity:** “Information-First Sampling” is not a standard terminology. A brief justification would improve clarity.

4) **Standard deviations missing:** No variance metrics are given. Please include standard deviations for a representative subset of experiments.

5) Missing related work on long-form video understanding: [B]

Thought experiment: Do you think long-form video understanding is fundamentally a long-context modelling problem?

==

[A] Chandrasegaran, Keshigeyan, et al. "Hourvideo: 1-hour video-language understanding." Advances in Neural Information Processing Systems 37 (2024): 53168-53197.

[B] Ye, Jinhui, et al. "Re-thinking temporal search for long-form video understanding." Proceedings of the Computer Vision and Pattern Recognition Conference. 2025.

---

> ### Author Rebuttal · Authors · 2025-07-31
>
> ## About reproducibility & release plan
>
> The model weight and inference code are open to the public now. The training code and dataset are under legal review. They will be released when the legal review is done.
>
> ## About the clarity of information-first sampling
>
> In brief, **Information-First Sampling is a dynamic token-allocation strategy that, under a fixed context budget, selects the most informative text spans and image regions so that critical content remains intact.** It has two complementary components:
>
> Image Area Preservation (IAP) – fully preserves spatial image information.
>
> Automatic Degradation Sampling (ADS) – ensures temporal inputs and their supervision signals are retained in full.
>
> Together, ADS and IAP let the model devote its limited tokens to the meaningful patches.
>
> ## About standard deviations missing
>
> Existing closed-source model APIs enable sampling during inference, which introduces uncertainty and can lead to bias in multiple tests. **However, for open-source VLMs, the common practice is to directly use greedy decoding.** Our experiments are run with decoding temperature τ = 0 that is common practice, thus we employ greedy decoding for every evaluation. Because this setup is fully deterministic, repeated evaluations yield identical predictions and standard deviations are not applicable.
>
> In addition, we fix **the same random seed throughout the entire training pipeline**—covering Python’s `random`, NumPy, PyTorch, and data-loader workers—so that initialization, data shuffling, and gradient updates are fully reproducible.
>
> We will make these settings explicit in the revised manuscript to enhance transparency and avoid any potential confusion.
>
> ## About the missing related work
>
> We acknowledge that the current draft requires more comprehensive coverage of prior work in this domain and will address this in the final manuscript.
>
> ## Do you think long-form video understanding is fundamentally a long-context modelling problem?
>
> Not entirely. For images, visual understanding can largely be regarded as a short-context spatial modeling task (e.g., modeling objects, scenes, and spatial relationships). Videos, however, introduce the temporal dimension. As a result, video understanding requires both spatial modeling of individual frames and temporal modeling across frames (e.g., modeling events, motion, and changes over time). Because the temporal span can grow without bound, understanding long-form videos within **general** multimodal large language models ultimately manifests as a long-context modeling problem. Moreover, the information density of video is typically much lower than that of text, leading to substantial variability in sparsity when processed by general-purpose models. Various sparsity-oriented methods—some designed specifically for video and others applicable to long-context modeling more broadly—can be leveraged to address this issue. Exploring these directions is an important avenue for future work, though it lies beyond the scope of this paper.

---

> > ### Comment · Reviewer_4qXs · 2025-08-06
> > **Reviewer Response**
> >
> > Thank you authors for the rebuttal. Authors have addressed my concerns.
> >
> > I will maintain my positive rating.

---

> ### Author Response · Authors · 2025-08-01
> **add the missing results of hourvideo breakdown**
>
> We found that the results of hourvideo breakdown ware missing in the **second** submission. Here we append the results:
>
> ## **About HourVideo breakdown**
>
> Here is the per-category results for HourVideo:
>
> | **category** | **Dev** | **Test** |
> | --- | --- | --- |
> | navigation/object_retrieval | 30.7692 | 26.0417 |
> | navigation/object_retrieval_image | 30.7692 | 28.125 |
> | navigation/room_to_room | 28.5714 | 28.3333 |
> | navigation/room_to_room_image | 57.1429 | 20 |
> | perception/information_retrieval/factual_recall | 49.6 | 48.8898 |
> | perception/information_retrieval/sequence_recall | 37.5 | 39.5785 |
> | perception/information_retrieval/temporal_distance | 38.7097 | 25.8427 |
> | perception/tracking | 50 | 46.9274 |
> | reasoning/causal | 40 | 39.3333 |
> | reasoning/predictive | 62.5 | 58.7224 |
> | reasoning/spatial/layout | 20 | 15.5556 |
> | reasoning/spatial/proximity | 46.1538 | 34.6247 |
> | reasoning/spatial/relationship | 35.5556 | 33.2451 |
> | reasoning/temporal/duration | 47.5 | 40.9254 |
> | reasoning/temporal/frequency | 40.411 | 39.8347 |
> | reasoning/temporal/prerequisites | 57.6271 | 54.5113 |
> | summarization/key_events_objects | 42.1053 | 59.5289 |
> | summarization/temporal_sequencing | 60 | 60.5263 |
> | reasoning/counterfactual | 38.0952 | 34.4371 |
> | summarization/compare_and_contrast | 70 | 55.7895 |
> | Overall | 44.5008 | 41.8388 |

---

> ### Author Response · Authors · 2025-08-05
>
> Thank you again for your thoughtful review. We would appreciate it if you could let us know whether our rebuttal has adequately addressed your concerns. If there are any remaining issues, we would be glad to further clarify.

---

### Official Review · Reviewer_36zH · 2025-07-06

**Clarity:** 2
**Significance:** 2
**Originality:** 3
**Rating:** 4
**Confidence:** 3

**Summary:**

This paper proposes Eagle 2.5, a vision-language model (VLM) designed for long-context multimodal understanding. It systematically studies training strategies and data recipes for this setting, introducing methods such as information-first sampling with image area preservation and automatic degradation sampling, as well as progressive mixed post-training to enhance long-context capabilities. Additionally, the authors curate Eagle-Video-110K, a novel long-video dataset with combined story-level and clip-level annotations. Experiments show that Eagle 2.5 achieves competitive performance across a range of image and video understanding benchmarks.

**Questions:**

Please check the questions in the weaknesses above.

**Ethical Concerns:**

["NO or VERY MINOR ethics concerns only"]

**Final Justification:**

The authors’ rebuttal has addressed my concerns, and I will maintain my positive rating for the paper.

**Limitations:**

Yes

**Quality:**

3

**Strengths And Weaknesses:**

Strengths:

1. The paper provides a detailed description of the training strategies and data recipe for building an effective framework for long-context VLMs, which will benefit future community work in this area.

2. The proposed Eagle 2.5 framework achieves strong results, demonstrating competitive performance across a variety of image and video understanding benchmarks.

3. The paper includes detailed ablation studies that validate the effectiveness of the proposed components, studying the impact of elements such as information-first sampling, post-training schedules, and the Eagle-Video-110K dataset.

Weaknesses:

1. While the data and training strategies are well-executed, the core model architecture follows existing VLM designs. The novelty of the work is relatively limited.

2. Continuing from the previous point, the paper does not appear to mention what base model Eagle2.5-8B is built upon, and there is also no baseline performance reported for this base model in the experiments. This makes it difficult to assess what really contributes to the superior performance of Eagle, especially since the paper discusses many engineering efforts, which feels a bit messy without a clear baseline for reference.

3. Based on Table 7, it appears that training with the Eagle-Video-110K dataset brings only marginal improvements for long-video benchmarks (comparing the first and third rows). In Figure 6, the improvements from Eagle-Video-110K under the 64K context length setting also appear to be quite limited. It would be helpful if the authors could help understand this observation and clarify under what conditions Eagle-Video-110K contributes more.

---

> ### Author Rebuttal · Authors · 2025-07-31
>
> ## About the novelty of this work
>
> The success of large-scale multimodal models stems from a combination of factors, including—but not limited to—model design, large-scale multimodal data, and effective training strategies. In this work, we focus on these holistic innovations and improvements. Specifically, we treat **data sampling and tiling methods** as part of model preprocessing, as they are distinct from the raw data itself or the training method. As noted by both Reviewer 4qXs and Reviewer YT9W, our contributions also include **a novel training strategy and a valuable video dataset** enriched with dual annotations, which together enhance the capabilities and robustness of models tackling the challenging task of long-form video understanding.
>
> ## About the base model
>
> Eagle2.5-8B is **built on SigLIP2-400M and Qwen2.5-7B, without relying on any existing VLM base model.** Compared with Eagle2, which has three training stages (Stage 1, Stage 1.5, Stage 2), Eagle2.5 adopts five stages (Stage 1, Stage 1.5, Stage 2, Stage 3, Stage 4). The first two stages are consistent between the two models, focusing on pretraining with large-scale image–language data.
>
> In Eagle2, Stage 2 trains the model on high-quality image-language data. In contrast, Stages 2–4 of Eagle2.5 progressively train the model on long-context data. A slight difference exists in Stage 1.5, resulting in Eagle2.5-Stage 1.5 achieving better performance than Eagle2-Stage 1.5 and Eagle2-stage2.
>
> To isolate the impact of our improvements, we present the following key benchmark comparisons.
>
> | Benchmark | Eagle 2 (Stage2) | Eagle 2.5-Stage1.5 | Eagle 2.5 Final |
> | --- | --- | --- | --- |
> | DocVQA | 92.6 | 92.6 | **94.1** |
> | ChartQA | 86.4 | **88.3** | 87.5 |
> | InfoQA | 77.2 | 78.8 | **80.4** |
> | TextVQA | 83.0 | **84.6** | 83.7 |
> | OCRBench | 868 | 868 | **869** |
> | MMstar | 62.6 | **66.5** | 66.2 |
> | RWQA | 69.3 | 74.4 | **76.7** |
> | AI2D | 83.9 | **85.5** | 84.5 |
> | MMMU | **56.1** | 54.0 | 55.8 |
> | HallB | 49.3 | 53.4 | **54.7** |
> | MathVista | 63.8 | 65.1 | **67.8** |
> | Video-MME (16 frames) | 56.6 | 58.9 | **65.0** |
> | Video-MME (128 frames) | 58.7 | 61.1 | **70.8** |
>
> These results show that Eagle2.5-8B substantially outperforms Eagle 2 and Eagle2.5-stage1.5 model on video understanding tasks, while maintaining competitive performance on image-language tasks. This confirms that the introduced extensions are effective and generalizable.
>
> ## Performance Improvement of Eagle-Video-110K
>
> Overall, incorporating Eagle-Video-110K yields only a modest overall improvement because gains on long-form videos are partially counterbalanced by small drops on short clips, in some video benchmarks, especially Video-MME.
>
> Video-MME includes three subsets: long, medium, and short. Therefore, when adding long video data to the original data, the evaluation results on different subsets will be offset. This will lead to limited improvement in overall performance. For example, on the setting in Table 7 (testing up to 64 frames), the performance of different subsets with or without eagle-video-100k is as follows:
>
> | Model | Long | Medium | Short | Total |
> | --- | --- | --- | --- | --- |
> | 32K→64K, Open-Data | 59.5 | 63.4 | 81.3 | 68.1 |
> | 32K→64K, Open-Data + Eagle-Video-110K | 61.8 | 64.6 | 80.0 | 68.8 |
> | gain | **+2.3** | **+1.2** | **-1.3** | **+0.7** |
>
> Adding Eagle-Video-110K yields a sizeable +2.3-point gain on the *long* subset (and +1.2 on *medium*), but a –1.3-point drop on *short*, translating to a modest +0.7 overall. These findings suggest that jointly tuning for both long- and short-context videos remains an open, worthwhile direction for future work.

---

> > ### Comment · Reviewer_36zH · 2025-08-05
> >
> > Thank you for the clarification and additional experiments, which have addressed my concerns.

---

> ### Author Response · Authors · 2025-08-05
>
> Thank you again for your thoughtful review. We would appreciate it if you could let us know whether our rebuttal has adequately addressed your concerns. If there are any remaining issues, we would be glad to further clarify.

---

### Decision · Program_Chairs · 2025-09-17

**Decision:**

Accept (poster)

**Comment:**

The paper introduces Eagle 2.5, an 8 B-parameter vision-language model (VLM) aimed at native long-context reasoning across both video and high-resolution image domains. Key contributions recognized by reviewers are: Information-First Sampling (IFS) strategy, progressive training recipe, and a 110K clip dataset named Eagle-Video-110K.

All reviewers agree that this work tackles a highly relevant and challenging frontier in multimodal AI: long-context visual understanding. The proposed data curation and training strategies are well-motivated and shown to be effective. The proposed 8B model achieves strong empirical results, and notably outperforms Gemini 1.5 on the challenging HourVideo benchmark.

Two reviewers raised concerns about reproducibility and release plan of the work, and asked for some detailed quantitative & qualitative results. The authors addressed these concerns in the rebuttal. As a result, all four reviewers reached a consensus to accept/borderline accept the paper.